# Study of Alternative Imaging Methods for In Vivo Boron Neutron Capture Therapy

**DOI:** 10.3390/cancers15143582

**Published:** 2023-07-12

**Authors:** Dayron Ramos López, Gabriella Maria Incoronata Pugliese, Giuseppe Iaselli, Nicola Amoroso, Chunhui Gong, Valeria Pascali, Saverio Altieri, Nicoletta Protti

**Affiliations:** 1Dipartimento Interateneo di Fisica, Università degli Studi di Bari Aldo Moro, 70125 Bari, Italy; gabriella.pugliese@ba.infn.it (G.M.I.P.); giuseppe.iaselli@ba.infn.it (G.I.); 2Istituto Nazionale di Fisica Nucleare, Sezione di Bari, 70125 Bari, Italy; nicola.amoroso@uniba.it; 3Dipartimento di Farmacia-Scienze del Farmaco, Università degli Studi di Bari Aldo Moro, 70125 Bari, Italy; 4School of Environmental and Biological Engineering, Nanjing University of Science and Technology, Nanjing 210094, China; gongchunhui@njust.edu.cn; 5Istituto Nazionale di Fisica Nucleare, Sezione di Pavia, 27100 Pavia, Italy; valeria.pascali01@universitadipavia.it (V.P.); saverio.altieri@unipv.it (S.A.); nicoletta.protti@unipv.it (N.P.); 6Dipartimento di Fisica, Università degli Studi di Pavia, 27100 Pavia, Italy

**Keywords:** Monte Carlo, BNCT, imaging, Compton camera, MLEM, deep learning

## Abstract

**Simple Summary:**

Accurate in vivo boron dosimetry is crucial for successfully implementing Boron Neutron Capture Therapy in clinical settings. This investigation uses a Compton camera detector and Monte Carlo algorithms to evaluate different imaging methods for dosimetry and tumor monitoring. The study employed the Maximum Likelihood Expectation Maximization method for dosimetry tomography, while morphological filtering and Deep Learning techniques with Convolutional Neural Networks were explored for tumor monitoring. The research is significant as it addresses the critical need for precise in vivo boron dosimetry in Boron Neutron Capture Therapy. The findings highlight the potential of the Maximum Likelihood Expectation Maximization method for accurately assessing the boron dose and demonstrate the promising results of the Convolutional-Neural-Network-based approach for tumor monitoring. The research emphasizes the importance of optimizing imaging methods and clinical parameters in this treatment, paving the way for improved treatment outcomes and enhanced patient care.

**Abstract:**

Boron Neutron Capture Therapy (BNCT) is an innovative and highly selective treatment against cancer. Nowadays, in vivo boron dosimetry is an important method to carry out such therapy in clinical environments. In this work, different imaging methods were tested for dosimetry and tumor monitoring in BNCT based on a Compton camera detector. A dedicated dataset was generated through Monte Carlo tools to study the imaging capabilities. We first applied the Maximum Likelihood Expectation Maximization (MLEM) iterative method to study dosimetry tomography. As well, two methods based on morphological filtering and deep learning techniques with Convolutional Neural Networks (CNN), respectively, were studied for tumor monitoring. Furthermore, clinical aspects such as the dependence on the boron concentration ratio in image reconstruction and the stretching effect along the detector position axis were analyzed. A simulated spherical gamma source was studied in several conditions (different detector distances and boron concentration ratios) using MLEM. This approach proved the possibility of monitoring the boron dose. Tumor monitoring using the CNN method shows promising results that could be enhanced by increasing the training dataset.

## 1. Introduction

Boron Neutron Capture Therapy (BNCT) is an innovative hadrontherapy with high selectivity over cancer tissue based on the neutron capture reaction ${}^{10}B{\left(n,\alpha \right)}^{7}Li$ [1,2]. BNCT is performed by first delivering a specifically designed ${}^{10}B$-enriched molecule to the tumor cell and by the subsequent irradiation with thermal neutrons. The non-radioactive isotope boron-10, when irradiated with low energy neutrons (cross section of 3837 barn at 0.025 eV), produces a boron-11 unstable nucleus which converts in approximately 10${}^{-12}$ s with a 94% yield into an α-particle and an excited ${}^{7}Li$ recoil nucleus. The excited lithium nucleus then decays, emitting a 478 keV gamma ray (prompt gamma). The reaction has a positive Q-value of 2.78 MeV and the products have a high linear energy transfer (LET) that deposits the energy within a range of <10 μm, making energy transfer possible (up to 83% on average) in the cancer cell.

A topic of primary importance in BNCT dosimetry is the measurement of the dose given to the patient during treatments. Nowadays, boron dosimetry is carried out through indirect and separated measurements of boron-10 concentrations by several blood tests (before, meanwhile and after irradiation) and local thermal neutron flux estimations made by treatment planning systems. This procedure leads to large uncertainties in clinical dosimetry. Therefore, a possible protocol based on the measurement of the prompt gamma-ray of 478 keV becomes significant to be able to reconstruct in vivo the spatial distribution of the dose in the area of interest. To such an aim, we will investigate imaging algorithm methods such as computed tomography using “Compton camera techniques” [3].

In this approach, a 3D sensor capable of determining, in addition to the energy of the emitted photon, its position is needed. In this context, a high-resolution 3D cadmium–zinc–telluride (CZT) drift strip detector has been proposed, able to perform room temperature measurements of photon energy, timing and 3D positioning up to the MeV region. Recently, very preliminary results of the spectroscopic capabilities of such a spectro-imager have been reported in the energy range of interest of BNCT [4].

In this work, we have used Monte Carlo methods to simulate a Compton image dataset within a BNCT approach in order to study imaging methods. Three methods were employed. The classic Maximum Likelihood Expectation Maximization (MLEM) iterative method was analyzed for dose reconstruction, while for tumor monitoring, two approaches were tested: a method based on image processing operations and another applying segmentation with Convolutional Neural Networks (CNNs). All methods can be found explained in detail below.

## 2. Compton Imaging

The Compton camera method to reconstruct images is based on Compton kinematics to find the origin of the detected gamma ray. Typically, the camera is constituted by two distinct detectors: the first one is known as a scatterer and has the function of promoting the Compton interaction of the incident gamma and the second one is known as the absorber to stop, by the photoelectric effect, the scattered photon. Generally, the two detectors are separated, thus delimiting a small working solid angle of the Compton camera. The possibility of having a double layer detector acting at the same time as the scatterer as well as the absorber improves the detection efficiency of the system [5].

The hits in the two detectors and the measured absorbed energies are used to calculate the scattering angle, as is shown in Figure 1a, and the most likely origin spot by superposition of different Compton cones, as represented in Figure 1b. In the present study, we take into account only the events in which a photon has Compton scattering and photoabsorption in two different detector planes. Such events are called *true events*. In this sense, the development of a 3D spectro imager allows 4π Compton imaging, solving the Compton kinematic equation. Then, knowing the gamma incident energy, the measurement of the absorbed energy fully determines the Compton scattering angle according to:(1)$$\begin{array}{ccc} \cos\theta & = & 1-m_{e}c^{2}\left(\frac{1}{E_{2}}-\frac{1}{E_{\gamma }}\right);\\ E_{\gamma } & = & E_{1}+E_{2} \end{array}$$
where $E_{\gamma }$ is the incident photon energy and $E_{1}$ and $E_{2}$ are the energies deposited in the scatter layer and absorption layer, respectively.

The image revealed directly by the detector using Compton kinematics will be referred as the “back-projection image” in what follows.

## 3. Dataset

A set of Monte Carlo simulated data was generated using the GEANT4 framework [6]. The code was exploited in the 10.7 software version. In order to implement the most accurate low-energy models, the Electromagnetic Standard Option 4 (emstandard_opt4) physics list package was used.

The simulated setup was composed of a solid cylinder phantom made of air or soft tissue (both defined in GEANT4 material database) with a 30 mm radius and a 100 mm height, and a Compton camera with two adjacent CZT crystal layers each with dimensions of 20 × 20 × 5 mm${}^{3}$ and a 5.78 g cm${}^{-3}$ density. The phantom radius was chosen by approximating a mouse model [7]. The detector was situated in the *Z*-axis as shown in Figure 2. The distance between the detector and the gamma source (yellow mark) in the figure can be varied.

The source gamma energy was set at 478 keV and two geometric configurations were simulated: five vertices source with point-like emitters (6 mm apart) and a single spherical source with a 10 mm radius.

In both configurations, the source was centered at the coordinates system origin. Since in realistic BNCT treatments, boron can load also healthy tissue in a given proportion, we define the tumor-to-healthy boron concentration ratio T/N as the boron concentration between tumor tissue and healthy tissue. Additional random gammas were generated in the phantom with vertex positions outside the source with given T/N values.

The hits position (without taking into account the uncertainties introduced by the intrinsic space resolution of the detector) and the energy deposited of *true events* were used to reconstruct the back-projection images.

These images were set in gray-scale, normalized to a maximum pixel intensity, and provided in .csv format structured in eight fields: pixel intensity; x, y, and z pixel position; i, j, and k pixel indexes, and a binary source which corresponds to 0 or 1 related to the simulated source.

## 4. Iterative Method for Dose Monitoring

At first, we employed a list mode MLEM (LM-MLEM) method, described in [8,9] and already proven to be suitable for Compton imaging. The iterative algorithm is given by:(2)$$\lambda_{j}^{\left(n+1\right)}=\frac{\lambda_{j}^{\left(n\right)}}{s_{j}}\sum_{i=1}^{M}\frac{t_{ij}}{\sum_{k}t_{ik}\lambda_{k}^{\left(n\right)}}$$
where $\lambda_{j}^{\left(n\right)}$ is the value of pixel j at iteration n, $s_{j}$ is the sensitivity matrix representing the probability of a gamma ray emitted from pixel *j* to be detected, *M* is the number of measured events and the system matrix $t_{ij}$ is the probability of a gamma ray emitted from pixel *j* being detected by the measurement with index *i*.

The algorithm used for the calculation was a GPU-based image reconstruction code [10]. The images were reconstructed in a 100${}^{3}$ matrix pixels grid in the space volume from −60 mm to 60 mm for each axis (cube pixel image with 1.2 mm side) covering the entire phantom.

## 5. Methods for Tumor Monitoring

### 5.1. Morphological Image Processing

Morphological filters are mathematical operations employed in image processing to extract specific components from an image. In the context of image segmentation, erosion and dilation processes are commonly utilized as morphological filters.

Morphological dilation enhances the visibility of objects and fills in small holes within them. By expanding the boundaries of image components, it aids in consolidating fragmented regions and creating more complete and cohesive objects. This process effectively increases the size of objects while preserving their overall shape and structure.

On the other hand, erosion procedures implemented as part of morphological filters remove isolated or floating pixels, as well as thin lines. This operation reduces the background regions present in the image, making them smaller, while simultaneously expanding the regions belonging to foreground objects. As a result, the foreground objects become more pronounced and distinct from the background.

By applying both dilation and erosion operations, morphological filters offer a means to refine and improve image segmentation. These processes help to eliminate noise, connect fragmented regions, and emphasize the target objects of interest, ultimately facilitating more accurate and reliable segmentation results. These operations, as described in [11], have been applied as a supervised alternative method for segmenting Compton back-projection images.

### 5.2. Segmentation with CNN

A CNN is a network architecture for deep learning inspired by the perception mechanism of the natural vision of living creatures. The first prototype of a convolutional neural network dates back to 1980 [12]. To date, following numerous improvements, CNNs have been used in applications involving the recognition of images and objects, audio data, and signals. The use of this type of network eliminates the need to manually extract the features of interest, resulting in high precision recognition for larger datasets. For these reasons, CNNs are particularly suitable for solving biomedical imaging problems, where the precision and accuracy required for segmentation are particularly high.

Given that the dataset available was restricted to the simulated configurations, a data augmentation tool was used in order to increase the effectiveness of the classification model. This tool performs geometric transformations that allow to obtain a greater number of images. In addition, because of not having available a sufficient quantity of images to carry out training from scratch (limited data to simulated distribution sources), it was decided to apply an already trained model using the transfer learning technique.

Transfer learning consists of an advanced machine learning method in which a model, pre-trained to carry out a specific activity, is reused as a starting point for the development of a model intended for the execution of a new activity similar to the previous one. By reusing most of the parameters of the already trained CNN, the training phase is limited to the layer used for the classification or regression of the characteristics obtained through the pre-existing levels [13,14]. To carry out this operation, the first step is to obtain the layers belonging to the previously trained model, and to freeze them in order to avoid the loss of useful information in subsequent phases. Secondly, additional layers are added for the classification of the new dataset on the basis of the features already learned. A further optional step is represented by fine-tuning, i.e., the thawing of the entire model in order to re-train it on the new data using a very low learning rate. This can potentially significantly improve network efficiency.

To carry out this analysis, a DeepLab model was used, in particular DeepLabV3+, using ImageNet’s ResNet50 to extract features, both available in MATLAB. Deep Residual Nets (RESNET) [15] is a standard tool for the community, proposed for the classification of images. In this analysis, the feature map generated by ResNet50 as a DCNN model is input to the DeepLab model to perform the semantic segmentation.

After image pre-processing and dataset optimization, the network was trained with stochastic gradient descent with momentum (sdgm) [16]. The resulting network is a DAGNetwork object, with 206×1 Layers, 227×2 tables, and 1 cell as Input and Output. The acronym DAG replaces the expression *Directed Acyclic Graph* and indicates the peculiarity of the network architecture of having the structure of a direct acyclic graph.

### 5.3. Segmentation Validation

In order to quantify the segmentation goodness, some parameters were calculated. Since the images are digital, the most immediate way to quantify regions is to establish how many pixels belong to them.

The segmented images were binarized: the number of pixels with an intensity of ‘1’ belong to the overlapping region between the real image and the segmented one represents the number of *true positives* (TP); the number of pixels belonging to image segmented but not to the real source represents the number of *false positives* (FP); and finally, the number of pixels belonging to the real source but not present in the segmented image represents the number of *false negatives* (FN). For the result discussion we define:(3)$$\mathrm{Accuracy}=\frac{TP}{TP+FP}$$
(4)$$\mathrm{Sensitivity}=\frac{TP}{TP+FN}$$

Both quantities were calculated and given in percentages. Sensitivity is a parameter that estimates how many pixels actually belonging to the source and have been recognized as such. Accuracy, on the other hand, estimates the precision of the segmentation made.

## 6. Results and Discussion of Dose Monitoring

Spherical distribution was chosen to study the iterative method performance since it is the closest model to a real tumor. We analyzed the spherical source in the following situations: (1) in air or tissue to study the dispersion in the phantom itself and (2) in tissue with different T/N values (T/N = 5.0 and 2.0) and with the detector at different distances (30.0, 60.0, and 100.0 mm). From clinical studies, it is known that for effective BNCT treatment, the boron concentration ratio should be ∼3.0 or more [17]; the case T/N = 2.0 has been analyzed as an extreme case and T/N = 5.0 was chosen to cover a wide range given recent clinical results which achieved a tumor-to-normal ratio of 4.125 [18].

### 6.1. Algorithm Validation

In order to check the performance of the iterative algorithm, 250 iterations were carried out in air with the detector 60 mm away from the source. The spherical volume was reconstructed as shown in Figure 3. When the iteration number is as high as 250, the algorithm develops a sort of granularity in the XY-image (see Figure 3b). It was found that 50 iterations were enough to obtain an image with acceptable resolution according to the algorithm convergence and avoid the granularity issue.

Figure 4 shows the XY-plane tomography image of the spherical source in tissue after 50 iterations. Comparing the results in the two cases (air vs. tissue), it is possible to conclude that the dispersion effect due to the tissue is not appreciable.

In the case of tissue presence, the X and Y axis profiles were also analyzed and are reported in Figure 5.

### 6.2. Stretching along Z-axis

Is also interesting to study the XZ tomography image. Figure 6 shows the XZ-plane tomography image for the spherical source in tissue after 250 iterations (Figure 6a), and the reconstruction profiles along the Z-direction (Figure 6b). The presence of a stretching effect along the Z-axis is evident even with a large number of iterations.

Previous studies have already demonstrated that this stretching effect arises at larger Compton scattering angles (θ) [19]. Then, to investigate the stretching behavior, measurements with the detector at distance values in the range from 30 to 100 mm were performed. The Z-axis profiles after 250 iterations were fitted with Gaussian distributions. Figure 7 shows the fitted sigmas vs. the detector distance values; therefore, longer stretching is associated with larger distances.

### 6.3. Dependence on the Boron Concentration Ratio

The therapeutic ratio dependence was investigated with the detector 60 mm apart from the source, and therapeutic ratio values T/N = 5.0 and 2.0. The results are shown in Figure 8 and Figure 9, respectively. Looking at the figures, a noise increase when the boron-10 concentration increases is evident in the healthy tissue. This result confirms the expected experimental behavior, since at lower T/N values there is more energy deposited in the healthy tissue with respect to the tumor region, which enhances the image background in the reconstruction process. Nonetheless, the source was reconstructed at least in the xy-plane even with the lowest concentration under study.

## 7. Results of Tumor Monitoring Methods

### 7.1. Filtering Method Results

Filtering was applied to the following cases: (1) a spherical source with radius 10 mm in air; (2) a spherical source with radius 10 mm in tissue and T/N = 2.0; (3) a spherical source with 10 mm radius in tissue and T/N = 5.0; and (4) a source with 5 point vertices in air. In all cases, the detector was positioned at 60 mm.

The four configurations were segmented using between two and three cycles of erosion and dilation, each depending on the geometric shape of the source and the boron concentration ratio present. The real source images overlapped with the reconstructed images are shown in Figure 10. The regions in yellow represent the intersection between the real source and the reconstructed one (*true positive*); the red regions represent the image portions belonging only to the real source (*false negative*); and the green regions represent the reconstructed region only (*false positive*).

The results for the accuracy and sensitivity values are summarized in Table 1 for the four cases studied.

Although the application of morphological filters has proven to be effective at segmentation, if larger datasets were available, the procedure would be inefficient. For each of the various cases, in fact, there have been variations in the order of application of the filters, in the structural element used, or in the input parameters. Therefore, each case must be studied separately and the ways in which the morphological filters are used must be deduced by human supervision knowing a priori the tumor geometry.

### 7.2. Segmentation Performance Using CNN

After the training phase, discussed in Section 5.2, the CNN segmentation method was applied to the cases analyzed in Section 7.1. Figure 11 shows the simulated binary source overlapping the respective segmentation result. In the images, there are three colors: green refers to the real source, pink characterizes the pixels that the network classifies as belonging to the source, and white represents the overlap between the two regions.

We again computed the accuracy and sensitivity for this method and the results are reported in Table 2.

In the case of the spherical source (cases 1–3), the model predictions are correlated with the tumor-to-healthy boron concentration ratio. In that sense, the accuracy is deprecated with the selectivity of the treatment. The segmentation results are most accurate with the spherical source in air and less precise with T/N = 2.0 (case 1 > case 3 > case 2). Overall, it can be concluded that the results obtained by means of the CNN procedure are worse than those obtained with morphological filters; however, one should stress that a CNN is a human unsupervised method able to be systematized.

In reality, given the amount of simulated data provided in training to the network, the CNN predictions are an excellent starting point since they are able to identify the position and morphology of the various sources. Therefore, by increasing the size of the training dataset, one expects a significant improvement in the CNN performance [20].

## 8. Conclusions

The Compton tomography in BNCT treatment was studied by Maximum Likelihood Expectation Maximization. Various experimental setups were employed to analyze the configuration of a spherical gamma source. The preliminary evidence is that this approach would be capable of evaluating the boron dose inside the patient. Moreover, the image reconstruction algorithm does not suffer significant dispersion effects by the healthy tissue.

The study of *Z*-axis profiles opens the possibility to estimate the distance between the source (tumor) and detector through the calculation of the sigma stretching. In addition, by studying the tumor-to-healthy boron concentration ratio dependence, it was found that the algorithm and the simulated detector are sensitive to the different T/N values within the clinical range.

Two alternative segmentation methods were explored for tumor monitoring: morphological filtering involving erosion and dilation and Convolutional Neural Networks (CNN). While the first technique effectively segmented the spherical source, it struggled with the configuration of the five-point-like sources. Conversely, the CNN model was trained using transfer learning techniques for back-projection image segmentation, yielding results that were slightly less accurate than those achieved with morphological filters but required no human supervision. Since these methods are proposed for tumor monitoring, one can conclude that morphological filtering is not suitable because of the method dependence on the a priori knowledge of the boron distribution. Therefore, it is proposed that the CNN model, which could be an effective solution when increasing the number of images in the dataset with promising results.

## Figures and Tables

**Figure 1 cancers-15-03582-f001:**
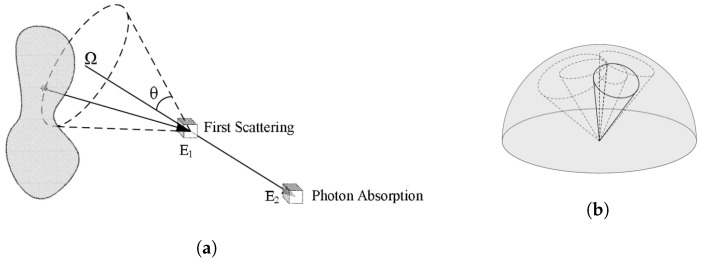
Compton camera imaging scheme. (**a**) Compton camera principle, (**b**) overlapped cones at the most likely position of the source.

**Figure 2 cancers-15-03582-f002:**
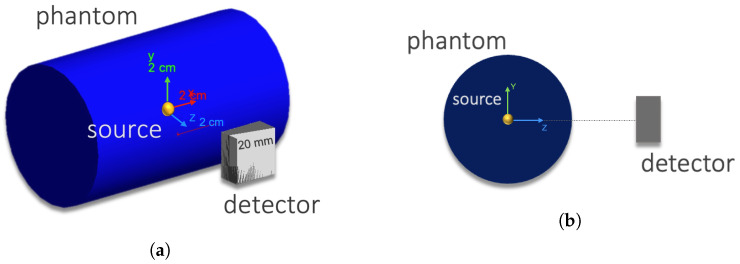
Simulated set-up in GEANT4. (**a**) General 3D view with the axes representing the reference scale. (**b**) View of the YZ plane.

**Figure 3 cancers-15-03582-f003:**
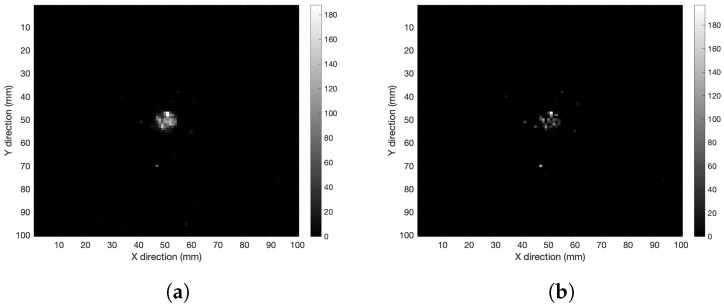
XY-plane tomography images of the spherical distribution in air after (**a**) 50 and (**b**) 250 iterations.

**Figure 4 cancers-15-03582-f004:**
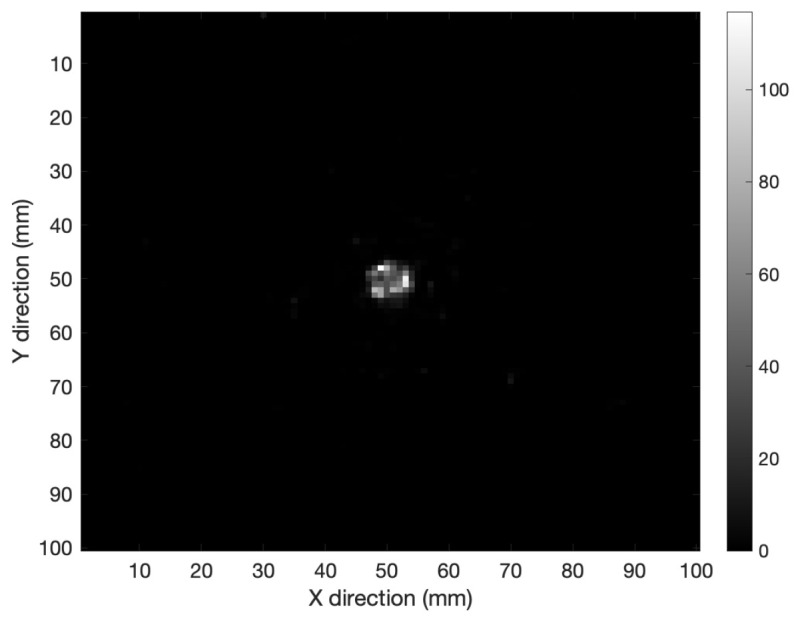
XY-plane tomography image for a spherical source in tissue after 50 iterations.

**Figure 5 cancers-15-03582-f005:**
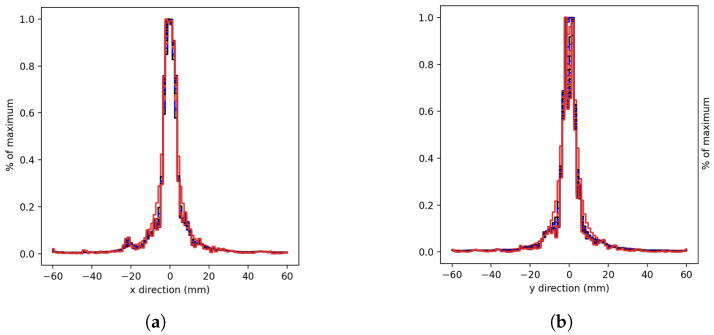
Spherical source normalized profiles in tissue. Each curve represents the profile with 10 more iterations until 250 iterations were reached. (**a**) X-axis, (**b**) Y-axis reconstruction.

**Figure 6 cancers-15-03582-f006:**
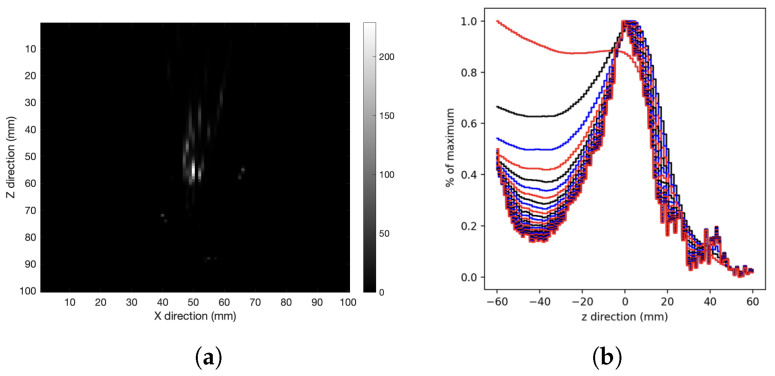
(**a**) XZ-plane tomography image, (**b**) Z-axis profile reconstruction for the spherical source in tissue after 250 iterations, each curve represents the profile with 10 more iterations.

**Figure 7 cancers-15-03582-f007:**
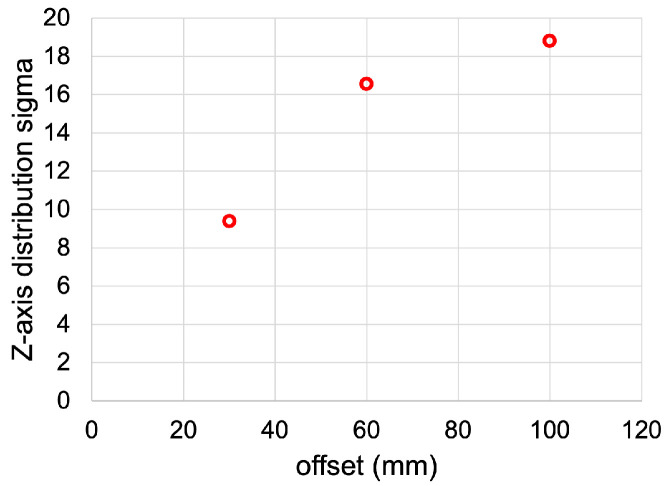
Gaussian distribution sigmas of Z-axis profiles for the spherical source at different detector distances.

**Figure 8 cancers-15-03582-f008:**
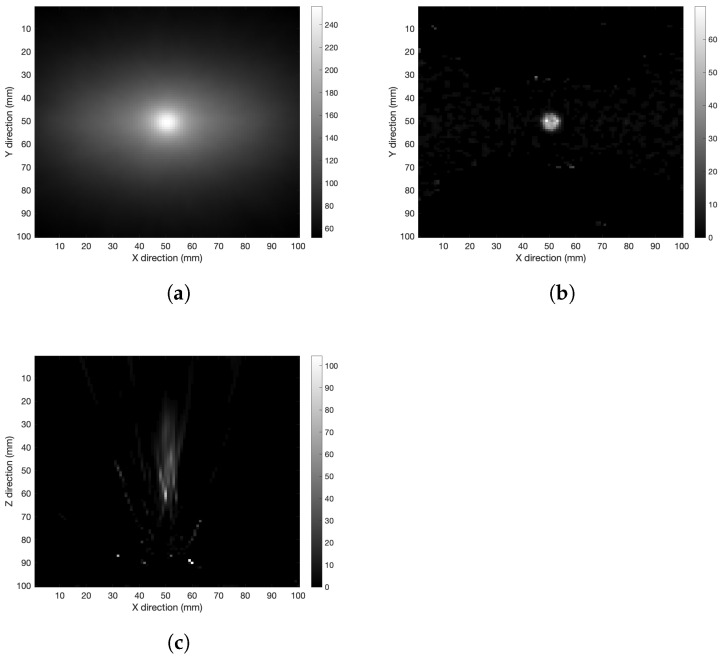
Spherical source with therapeutic ratio T/N = 5.0 and detector distance 60 mm. (**a**) XY-plane back-projection, (**b**) XY-plane tomography after 50 iterations, and (**c**) XZ-plane tomography after 250 iterations.

**Figure 9 cancers-15-03582-f009:**
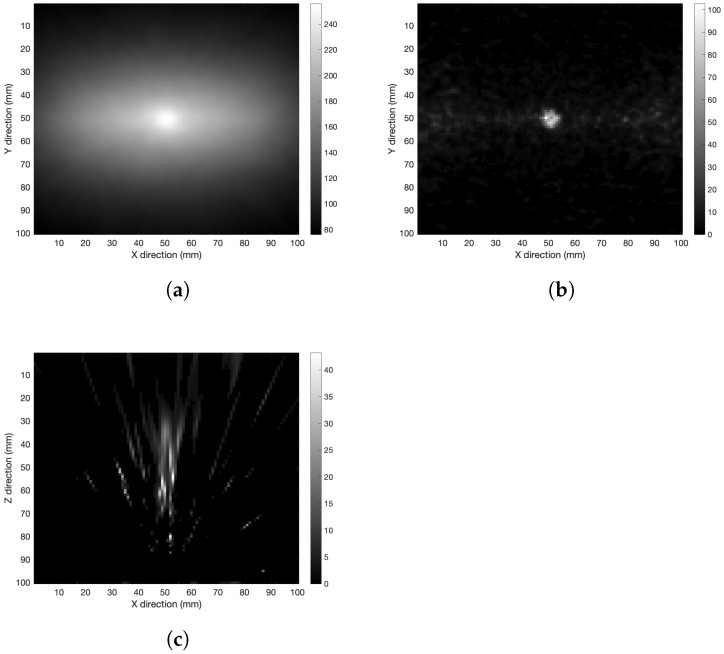
Spherical source with therapeutic ratio T/N = 2.0 and detector distance 60 mm. (**a**) XY-plane back-projection, (**b**) XY-plane tomography with 50 iterations, and (**c**) XZ-plane tomography with 250 iterations.

**Figure 10 cancers-15-03582-f010:**
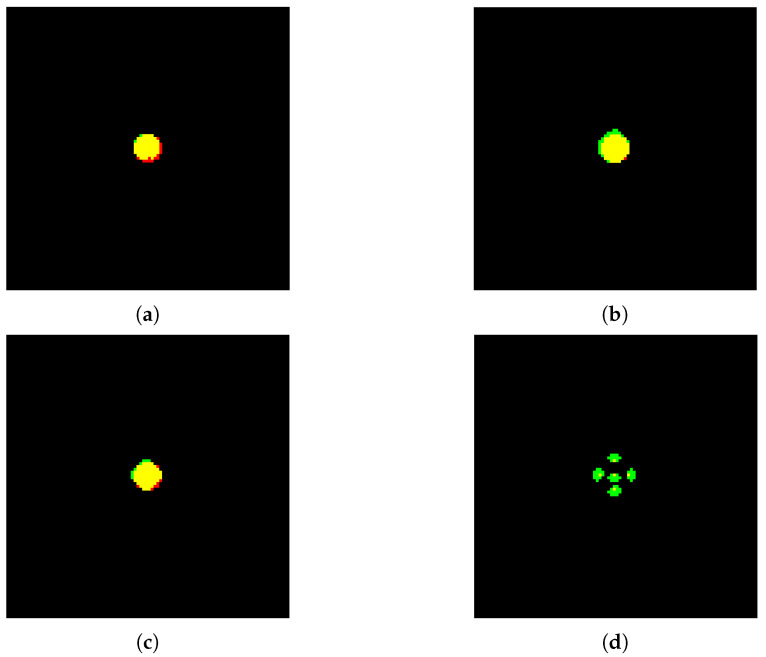
Segmentation using morphological filters: (**a**) case 1, (**b**) case 2, (**c**) case 3, and (**d**) case 4. Yellow regions: *true positive*, red regions: *false negative*, and green regions: *false positive*.

**Figure 11 cancers-15-03582-f011:**
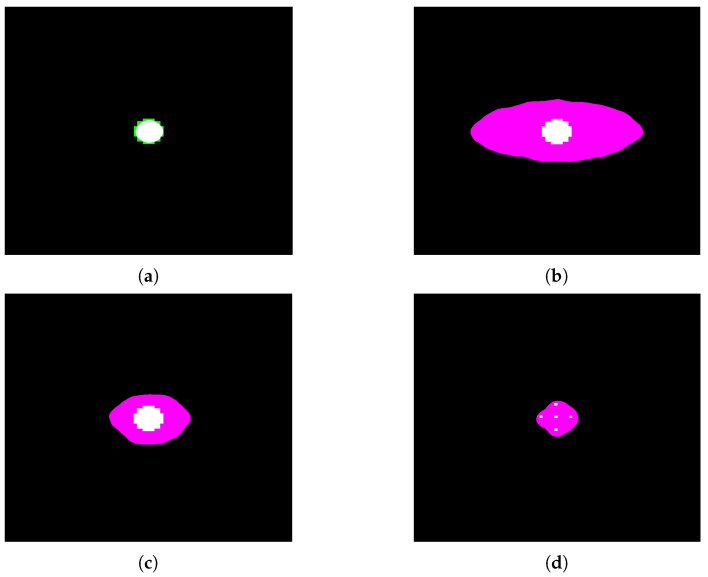
Segmentation using CNN: (**a**) case 1, (**b**) case 2, (**c**) case 3, and (**d**) case 4. Green regions: real source, pink: pixels classified by the network, and white: overlapping region.

**Table 1 cancers-15-03582-t001:** Reconstruction parameters using morphological filters.

	Accuracy	Sensitivity
case 1	97.0%	82.0%
case 2	79.0%	98.0%
case 3	90.0%	91.0%
case 4	8.0%	100%

**Table 2 cancers-15-03582-t002:** Parameters using the CNN approach.

	Accuracy	Sensitivity
case 1	79.56%	99.87%
case 2	4.83%	100%
case 3	19.68%	100%
case 4	3.81%	100%

## Data Availability

The data can be shared up on request.

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
