# Peer review of "Study of Alternative Imaging Methods for In Vivo Boron Neutron Capture Therapy"

_cancers, 2023, doi:10.3390/cancers15143582_

Round 1

Reviewer 1 Report

The article introduces the sophisticated challenges with the dosimetry of boron neutron capture therapy. This would be the near future therefore interest to readers maybe high. This manuscript is well established and would be useful for further studies and daily practices.

Author Response

Best regards, Dayron.

Reviewer 2 Report

This paper evaluates the potential use of a Compton camera for estimating BNCT dose using Monte Carlo simulations of a cylindrical phantom with spherically shaped tumor using GEANT4 software.  More specifically, the BNCT interactions were characterized as 478 keV gamma emissions due to Li decay.  The study analyzed the accuracy and sensitivity of a CZT detector to monitor the energy, positioning, and timing of BNCT emissions from a tumor with a 10-mm radius.

The background and technical concepts of the dosimetry application are well summarized in the manuscript.   To get a better understanding of the methods used, an unfamiliar reader would be required to read referenced papers.  The methods section of the manuscript needs stand on its own merit and improved so the reader will be able to understand what was done reading the text in the manuscript.

What is the justification for the phantom size selected?   What anatomical site is the 60-mm phantom radius intended to represent?    Is the BNCT dosimetry method intended primarily for superficial treatments?  Most tumors treated in radiation oncology are deep seated tumors.  The models used in this study do not seem to be clinically relevant.

The authors should Justify the TN rations of 5.0 and 2.0?  One of the challenges of BNCT is selectively concentrating sufficient 10B to achieve a therapeutic enhancement effect over conventional treatment without BNCT.  Are these values clinically relevant?

The authors should Justify the TN rations of 5.0 and 2.0?  One of the challenges of BNCT is selectively concentrating sufficient 10B to achieve a therapeutic enhancement effect over conventional treatment without BNCT.  Are these values clinically relevant?

Author Response

Best regards, Dayron.

Reviewer 3 Report

The paper “Study of alternative imaging methods for in-vivo Boron Neutron Capture Therapy” by Dayron Ramos Lopez et al. is quite interesting in that it reports different imaging methods tested for dosimetry and tumor monitoring in BNCT based on a Compton camera detector. A dedicated data-set was generated through Monte Carlo tools to study the imaging capabilities by using the Maximum Likelihood Expectation Maximization (MLEM) iterative method to study dosimetry tomography. As well, two methods based on morphological filtering and Convolutional Neural Networks (CNN) respectively, were used for the study of tumor monitoring. Furthermore, clinical aspects such as dependence by boron concentration ratio in the image reconstruction, and the stretching effect along the detector position axis were analyzed. The paper is worthy in in-vivo BNCT studies that it utilized non-invasive imaging methods to get the boron dosimetry and tumor monitoring information which is very crucial aspects for BNCT treatment. Segmentation validation and algorithm validation are analytically executed individually. In addition, the validation and verification of the study methods were performed with four different configurations including two therapeutic ratio values T/N 2.0 and 5.0. The comparison and discussion of the results are also depicted in conclusions.

The work is of a very good standard and can be published although I wonder if cancers is the right journal for propagation of this work.

I have few comments, suggestions, or questions as follows:

1. In the abstract, I see only the study motivation and the methods is mentioned. The description of research results or brief discussion should be added in to attract the attention of the readers on BNCT field. The keywords MLEM and Machine Learning are absent in abstract content.

2. In the Data-set, the simulated setup is clearly described that is composed a solid cylinder phantom made of air or soft tissue with 30 mm radius and 100 mm height, and a Compton camera with two adjacent CZT crystal layers with dimension 20x20x5 mm3 and 5.78 g cm3 density each. The camera was situated in the Z-axis. The distance between the detector and the gamma source (yellow mark) in the figure can be varied. The correlative diagram is depicted in Figure 2 the simulated set-up of GEANT4. I see no dimension and axis index labeled in this figure. It is not easy for the readers to understand the set-up of cylinder phantom and detector without a detailed figure. Furthermore, a detailed diagram can help the readers to figure out where is the right location of the distance between the detector and the gamma source. So, please revise Figure 2.

3. In Figure3~6, and in Figure 8~9, please check if their coordinates match with the corresponding location of Figure 2.

4. In Figure 7, the point at 200 mm had better be put in to make this check more comprehensive.

5. In 5.3. Segmentation validation, what is the objective to define the Accuracy and Sensitivity? Only for checking segmentation goodness? The highest values of them approach 1. Does that mean the higher values, the more accurate and sensitive of the results? What values would be acceptable? If the iteration results is not acceptable, how to improve? I would suggest the authors can explain more about this validation or indicate this with a flow chart.

6. The authors mention “In the case of spherical source with T/N = 5 the CNN predictions are very precise. On the other hand, the precision is quite poor in the rest of cases. Overall, it could be concluded that the results obtained by means of CNN procedure are worse than those obtained with morphological filters; however one should stress that CNN is a human unsupervised method. In reality, given the amount of simulated data provided in training to the network, the CNN predictions are an excellent starting point since they are able to identify the position and morphology of the various sources. Therefore, by increasing the number of data input to the network during its training phase, one expects the CNN performance to improve significantly.At the end of 7.2 Segmentation performance using CNN. But in section 7.1. Filtering method results spherical source with T/N = 5 is case 3. According to the Table 2, the accuracy of case 3 is 4.83% which is worse than case 1’s 79.56%. Please check the description. Moreover, the results obtained by means of CNN procedure are worse than those obtained with morphological filters. Is there any evidence that by increasing the number of data input to the network during its training phase, the CNN performance will be improved much?

No comments.

Author Response

Best regards, Dayron.

Round 2

Reviewer 3 Report

The authors have already revised the manuscript with great efforts and explain the questions according to the reviewer's comments.  I am satisfied with the answers and I think it will be very clear to the readers. Only some minor format problems should be carefully checked in fugure's or table's legends. I recommend the paper for publication.